# Unbounded Gradients in Federated Learning with Buffered Asynchronous Aggregation

**Mohammad Taha Toghani**
Rice University
Houston, TX, USA
mttoghani@rice.edu

**César A. Uribe**
Rice University
Houston, TX, USA
cauribe@rice.edu

## Abstract

Synchronous updates may compromise the efficiency of cross-device federated learning once the number of active clients increases. The *FedBuff* algorithm (Nguyen et al. [16]) alleviates this problem by allowing asynchronous updates (staleness), which enhances the scalability of training while preserving privacy via secure aggregation. We revisit the *FedBuff* algorithm for asynchronous federated learning and extend the existing analysis by removing the boundedness assumptions from the gradient norm. This paper presents a theoretical analysis of the convergence rate of this algorithm when heterogeneity in data, batch size, and delay are considered.

## 1 Introduction

Federated learning (FL) is an approach in machine learning theory and practice that allows training models on distributed data sources [12, 14]. The distributed structure of FL has numerous benefits over traditional centralized methods, including parallel computing, efficient storage, and improvements in data privacy. However, this framework also presents communication efficiency, data heterogeneity, and scalability challenges. Several works have been proposed to improve the performance of FL [1, 5, 7]. Existing works usually address a subset of these challenges while imposing additional constraints or limitations in other aspects. For example, the work in [8] shows a trade-off between privacy, communication efficiency, and accuracy gains for the distributed discrete Gaussian mechanism for FL with secure aggregation.

One of the most important advantages of FL is scalability. Training models on centralized data stored on a single server can be problematic when dealing with large amounts of data. Servers may be unable to handle the load, or clients might refuse to share their data with a third party. In FL, the data is distributed across many devices, potentially improving data privacy and computation scalability. However, this also presents some challenges. First, keeping the update mechanism synchronized across all devices may be very difficult when the number of clients is large [23]. Second, even if feasible, imposing synchronization results in huge (unnecessary) delays in the learning procedure [1]. Finally, each client often might have different data distributions, which can impact the convergence of algorithms [9, 21].

In synchronous FL, e.g., FedAvg [12, 14], the server first sends a copy of the current model to each client. The clients then train the model locally on their private data and send the model updates back to the server. The server then aggregates the client updates to produce a new shared model. The process is repeated for many rounds until the shared model converges to the desired accuracy. However, the existence of delays, message losses, and stragglers hinders the performance of distributed learning. Several works have been proposed to improve the scalability of federated/distributed learning via enabling asynchronous communications [1, 6, 11, 13, 15, 18, 20, 23]. In the majority of these results, each client immediately communicates the parameters to the server after applying a series of local

Table 1: Comparison of the characteristics considered in our analysis with relevant works for federated learning for **smooth & non-convex** objective functions. Parameter $\tau$ denotes the maximum delay.

| Algorithm | Reference | Asynchronous Update | Buffered Aggregation | Unbounded Gradient | Convergence Rate |
|---|---|:---:|:---:|:---:|:---:|
| FedAvg | McMahan et al. [14] | ✗ | ✓ | - | - |
| | Yu et al. [24] | ✗ | ✓ | ✗ | $\mathcal{O}\left(\frac{1}{\sqrt{T}}\right)$ |
| | Wang et al. [21] | ✗ | ✓ | ✓ | $\mathcal{O}\left(\frac{1}{\sqrt{T}}\right)$ |
| FedAsync | Xie et al. [23] | ✓ | ✗ | ✗ | $\mathcal{O}\left(\frac{1}{\sqrt{T}}\right) + \mathcal{O}\left(\frac{\tau^2}{T}\right)$ |
| FedBuff | Nguyen et al. [16] | ✓ | ✓ | ✗ | $\mathcal{O}\left(\frac{1}{\sqrt{T}}\right) + \mathcal{O}\left(\frac{\tau^2}{T}\right)$ |
| | This Work | ✓ | ✓ | ✓ | $\mathcal{O}\left(\frac{1}{\sqrt{T}}\right) + \mathcal{O}\left(\frac{\tau^2}{T}\right)$ |

updates. The server updates the global parameter once it receives any client update. This has the benefit of reducing the training time and better scalability in practice and theory [1, 6, 15, 17] since the server can start aggregating the client updates as soon as they are available.

The setup, known as "vanilla" asynchronous FL, has several challenges that must be addressed. First, due to the nature of asynchronous updates, the clients are supposed to deal with staleness, where the client updates are not up-to-date with the current model on the server [16]. Moreover, the asynchronous setup may imply potential risks for privacy due to the lack of secure aggregation, i.e., the immediate communication of every single client to the server [2, 3]. In [16], the authors proposed an algorithm called federated learning with buffered asynchronous aggregation (*FedBuff*), which modifies pure asynchronous FL by enabling secure aggregation while clients perform asynchronous updates. This novel method is considered a variant of asynchronous FL while serving as an intermediate approach between synchronous and asynchronous FL.

FedBuff [16] is shown to converge for the class of smooth and non-convex objective functions under the *boundedness* of the gradient norm. By removing this assumption, we provide a new analysis for FedBuff and improve the existing theory by extending it to a broader class of functions. We derive our bounds based on stochastic and heterogeneous variance and the maximum delay between downloads and uploads across all the clients. Table 1 summarizes the properties and rate of our analysis for FedBuff algorithm alongside and provides a comparison with existing analyses for FedAsync [23] and FedAvg [12, 14]. The rates reflect the complexity of the number of updates performed by the central server. The speed of asynchronous algorithms is faster since the constraint for synchronized updates is removed in asynchronous variations. To our knowledge, this is the first analysis for (a variant of) asynchronous federated learning with no boundedness assumption on the gradient norm.

## 2  Problem Setup & Algorithm

In this section, we first state the problem setup and then explain the FedBuff algorithm [16]. We consider a set of $n$ clients and one server, where each client $i \in [n]$ owns a private function $f_i : \mathbb{R}^d \to \mathbb{R}$ and the goal is to jointly minimize the average local cost functions via finding a $d$-dimensional parameter $w \in \mathbb{R}^d$ that

$$\min_{w \in \mathbb{R}^d} f(w) := \frac{1}{n} \sum_{i=1}^{n} f_i(w),$$
$$\text{with} \quad f_i(w) := \mathbb{E}_{\xi_i \sim p_i}[\ell_i(w, \xi_i)], \tag{1}$$

where $\ell_i : \mathbb{R}^d \times \mathcal{S}_i \to \mathbb{R}$ is a cost function that determines the prediction error of $w$ over a single data point $\xi_i \in \mathcal{S}_i$ on user $i$, and $p_i$ represents user $i$'s data distribution over $\mathcal{S}_i$, for $i \in [n]$. In the above definition, $f_i(\cdot)$ is the local cost function of client $i$, and $f(\cdot)$ denotes the global (average) cost function which the clients try to collaboratively minimize. Now, let $\mathcal{D}_i$ be a data batch sampled from

$p_i$. Similar to (1), we denote the stochastic cost function $\tilde{f}_i(w, \mathcal{D}_i)$ as follows:

$$\tilde{f}_i(w, \mathcal{D}_i) \coloneqq \frac{1}{|\mathcal{D}_i|} \sum_{\xi_i \in \mathcal{D}_i} \ell_i(w, \xi_i). \tag{2}$$

Minimization of (1) by having access to an oracle of samples and its variants are extensively studied for many different frameworks [7]. Now, we are ready to explain the FedBuff algorithm.

### 2.1    Federated Learning with Buffered Asynchronous Aggregation (FedBuff):

Let $w^0$ be the initialization parameter at the server. The ultimate goal is to minimize the cost function in (1), using an algorithm via access to the stochastic gradients. All clients can communicate with the server, and each client $i \in [n]$ communicates when its connection to the server is stable. First, let us explain the FedBuff algorithm from the client and server perspectives.

- **Client Algorithm**: Each client $i$ requests to read the server's parameter $w \in \mathbb{R}^d$ [1] once the connection is stable and the server is ready to send the parameter. There is often some delay in this step which we call the download delay. This may be originated from factors such as unstable connection, bandwidth limit, or communication failure. For example, maybe the server seeks to reduce the simultaneously active users by setting client $i$ on hold. The download delay can simply model all these factors. Once the parameter is received (downloaded) from the server, client $i$ performs $Q$ steps of local stochastic gradient descent starting from the downloaded model $w$ for its cost function $f_i(\cdot)$. In words, agent $i$ runs a $Q$-step algorithm (loop of size $Q$), where at each local round $q \in \{0, 1, \dots Q-1\}$, client $i$ samples a data batch $\mathcal{D}_{i,q}$ with respect to distribution $p_i$ and performs one step of gradient descent with local stepsize $\eta > 0$. Finally, agent $i$ returns the updates (the difference between the initial and final parameters) to the server. We refer to the time required to broadcast parameters to the server as the upload delay, which could have similar factors as the download delay. Agent repeats all this procedure until the server sends a termination message. Algorithm 1 summarizes the pseudo-code of operations at client $i \in [n]$, where Steps 4-8 show the local updates performed at the agent. Moreover, $\Delta_i$ in Step 9 denotes the difference communicated to the server.

---

**Algorithm 1** FedBuff (**Client** $i$)

---

1: **input:** number of local steps $Q$, local stepsize $\eta$.
2: **repeat**
3:     read $w$ from the server                                      ▷ download phase
4:     $w_{i,0} \leftarrow w$
5:     **for** $q = 0$ to $Q-1$ **do**
6:         sample a data batch $\mathcal{D}_{i,q}$
7:         $w_{i,q+1} \leftarrow w_{i,q} - \eta \nabla \tilde{f}_i(w_{i,q}, \mathcal{D}_{i,q})$
8:     **end for**
9:     $\Delta_i \leftarrow w_{i,0} - w_{i,Q}$
10:    client $i$ broadcasts $\Delta_i$ to the server                      ▷ upload phase
11: **until** not interrupted by the server

---

- **Server Algorithm**: The server considers an initialization for parameter $w^0 \in \mathbb{R}^d$. Then, starting from timestep $t = 0$, the server repeats an iterative procedure in addition to sending its parameters to the clients upon their request. Algorithm 2 describes the server operations in FedBuff. In a nutshell, the algorithm consists of two parts, (i) secure aggregation of client updates in a buffer with size $K \geq 1$ [2], and (ii) updating the parameters using the aggregated updates. In other words, let $k, t$ respectively denote the indices associated with buffer and server updates.[3] The server starting from $t = 0$, receives updates broadcast by the agents asynchronously depending on their

---

[1]For simplicity of presentation, we drop the timestep from the parameters at the client level. We will use the time notation in our analysis in Appendix A.

[2]$K$ is an integer number.

[3]As explained in [16], the buffer and secure aggregation may be performed on a secure channel which prevents the server from observing individual local updates received from the clients.

upload & download delays as well as the time required for $Q$ local updates. A secure buffered aggregates these updates, up to $K$ separate updates received by the clients in $\overline{\Delta}^0$, initially set to zero. By indexing $k$, we keep track of uploaded updates on the server. When the buffer saturates of $K$ different updates, server uses the aggregator parameter $\overline{\Delta}^0$ and updates its parameter $w^0$ according to line 9 of Algorithm 2. Then, the server increases its update counter $t$ and removes all updates from the buffer, i.e., $k = 0$. In this algorithm, we denote the agent which sends the $k$-th update at round $t$ by index $i_{t,k} \in [n]$. Basically, server repeats Steps 5-14 until some convergence criteria be satisfied. After the convergence, the server sends a termination message to all the clients.

---

**Algorithm 2** FedBuff (**Server**)

---

1: **input:** model $w^0$, server stepsize $\beta$, buffer size $K$
2: $t \leftarrow 0$, $k \leftarrow 0$
3: $\overline{\Delta}^0 \leftarrow 0$
4: **repeat**
5:     **if** the server receives an update $\Delta_{i_{t,k}}$ from some client $i_{t,k} \in [n]$ **then**
6:         $\overline{\Delta}^t \leftarrow \overline{\Delta}^t + \Delta_{i_{t,k}}$
7:         $k \leftarrow k + 1$
8:         **if** $k = K$ **then**
9:             $w^{t+1} \leftarrow w^t - \beta \overline{\Delta}^t$
10:            $k \leftarrow 0$
11:            $t \leftarrow t + 1$
12:            $\overline{\Delta}^t \leftarrow 0$
13:         **end if**
14:     **end if**
15: **until** not converged

---

As we described above, the crucial novelty of this algorithm is on the server side, where the server operations, with the help of a secure buffered aggregation, control the staleness and prevent unnecessary access to individual updates. Note that for $K = 1$, the presented algorithm reduces to vanilla asynchronous federated learning with no buffer aggregation. Figure 1 illustrates the update schedule for FedBuff and provides a comparison with the asynchronous updates in FedAvg [14]. As shown on the left of Figure 1, the vertical lines with light blue color are associated with uploaded updates. Note that the buffer size is $K = 2$ in this example. These vertical lines are of two types, (i) solid or (ii) hatched. The solid lines reflect the time the buffer is full, and hence the server performs an update. Contrary to FedBuff, under the synchronous updates (as shown in the right figure), the server should halt the training procedure until all clients selected within one round receive the updates.

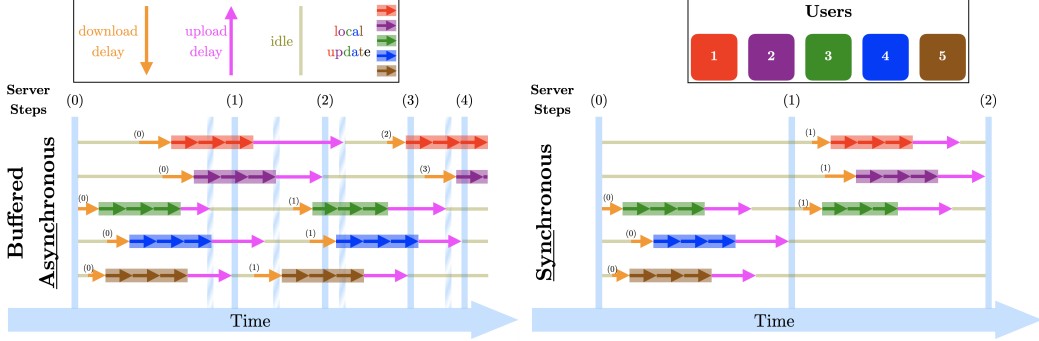

Figure 1: Communication and update schedule for synchronous and buffered asynchronous aggregation: The demonstrated setup in this example contains $n = 5$ agents, with $Q = 3$ local updates, buffer size $K = 2$ for FedBuff [16], and sampling rate 0.6 for FedAvg [14].

# 3 Convergence Result

This section presents our main result alongside a few standard assumptions. First, to be coherent with the proof in [16], let us denote $\tau_i^t$ to be the timestep of the last downloaded parameter on client $i \in [n]$ up to the $t$-th update at the server. Now, we are ready to introduce the assumptions in our analysis for FedBuff, i.e., Algorithms 1 & 2.

**Assumption 1** (Bounded Staleness). *For all clients, $i \in [n]$ and server steps $t \geq 0$, the staleness or effective delay between the download and upload steps is bounded by some constant $\tau$, i.e.,*

$$\sup_{t \geq 0} \max_{i \in [n]} \left| t - \tau_i^t \right| \leq \tau, \tag{3}$$

*and the server receives updates uniformly, i.e., $i_{t,k} \sim \mathrm{Uniform}([n])$.*

Note that $\tau_i^t$ is the timestep of the last parameter downloaded via agent $i$ up to timestep $t$ at the server. Therefore, if agent $i$ contributes in the $(t+1)$-th update, i.e., $i_{t,k} = i$, for some $k \in \{0, 1, \ldots, K-1\}$, the difference between the download and upload rounds is bounded. This is a standard assumption in the analysis of asynchronous algorithms with heterogeneous data on the clients.[4]

**Assumption 2** (Smoothness). *For all clients $i \in [n]$, function $f_i : \mathbb{R}^d \to \mathbb{R}$ is bounded below, differentiable, and $L$-smooth, i.e., for all $w, u \in \mathbb{R}^d$,*

$$\|\nabla f_i(w) - \nabla f_i(u)\| \leq L \|w - u\| \tag{4}$$

$$f_i^\star := \min_{w \in \mathbb{R}^d} f_i(w) > -\infty. \tag{5}$$

This assumption guarantees the necessary conditions for analyzing smooth & non-convex functions. Note that boundedness from below can be relaxed only to the global cost function $f$, i.e., it is sufficient to only assume that $f^\star := \min_{w \in \mathbb{R}^d} f(w) > -\infty$ in our analysis instead of (5) for all $i \in [n]$. Now, we introduce the assumptions on bounded stochasticity and heterogeneity.

**Assumption 3** (Bounded Variance). *For all clients $i \in [n]$, the variance of a stochastic gradient $\nabla \ell_i(w, \xi_i)$ on a single data point $\xi_i \in \mathcal{S}_i$ is bounded, i.e., for all $w \in \mathbb{R}^d$*

$$\mathbb{E}_{\xi_i \sim p_i} \|\nabla \ell_i(w, \xi_i) - \nabla f_i(w)\|^2 \leq \sigma^2. \tag{6}$$

This assumption is conventional in the analysis of stochastic optimization algorithms and has been used in many relevant works [1, 9–11, 16, 19, 21]. Note that as we defined the stochastic loss in (2) and used the stochastic gradients in Step 7, we also need to show the stochastic variance for the gradients of the sampled batches. For simplicity, let us assume that all batch sizes are of size at least $b$, therefore according to (6), we have:

$$\mathbb{E}_{p_i} \left\| \nabla \tilde{f}_i(w, \mathcal{D}_i) - \nabla f_i(w) \right\|^2 \leq \frac{\sigma^2}{|\mathcal{D}_i|} \leq \hat{\sigma}^2 := \frac{\sigma^2}{b}. \tag{7}$$

**Assumption 4** (Bounded Population Diversity). *For all $w \in \mathbb{R}^d$, the gradients of local functions $f_i(w)$ and the global function $f(w)$ satisfy the following property:*

$$\frac{1}{n} \sum_{i=1}^n \|\nabla f_i(w) - \nabla f(w)\|^2 \leq \gamma^2. \tag{8}$$

In our analysis, we work with heterogeneous cost functions. Therefore, it is a reasonable and conventional assumption to assume that the boundedness of the population diversity [4, 5, 16, 20]. The inequality in 9 measures the variance of local full gradients from the average full gradient, which resembles to the expressions in (6) & (7). Fallah et al. [5] discusses the connection of this bound to the similarity of local data distributions $p_i$, for all $i \in [n]$.

We now move to present our convergence result under the above assumptions.

---

[4]It is worth mentioning that Mishchenko et al. [15] relaxed this assumption (to unbounded delay) for the analysis of homogeneous smooth & strongly convex functions.

**Theorem 1.** *Let Assumptions 1-4 hold, $\beta = \frac{1}{K}$, and $\eta = \frac{1}{Q\sqrt{LT}}$. Then, the following property holds for the joint iterates of Algorithms 1 and 2: for any timestep $T \geq 160L(Q+7)(\tau+1)^3$ at the server*

$$\frac{1}{T}\sum_{t=0}^{T-1}\mathbb{E}\left\|\nabla f\left(w^t\right)\right\|^2 \leq \frac{8\sqrt{L}\left(f(w^0)-f^\star\right)}{\sqrt{T}} + \frac{16\sqrt{L}\left(\frac{\sigma^2}{b}+\gamma^2\right)}{\sqrt{T}}$$
$$+ \frac{320L(Q+1)(\tau^2+1)\left(\frac{\sigma^2}{b}+n\gamma^2\right)}{T}.$$

We present the proof for Theorem 1 in Appendix A.

The above theorem states the convergence of FedBuff algorithm to a first-order stationary point. This result states a convergence rate of $\mathcal{O}\left(\frac{1}{\sqrt{T}}\right) + \mathcal{O}\left(\frac{\tau^2}{T}\right)$, where the term affected by the maximum delay (second term) decays faster, hence the same convergence complexity as the synchronized counterpart. Note that this rate states the number of updates occurring on the server (iteration complexity), which in the case of asynchronous updates, practically converges much faster ($3.3\times$ according to [16]) than synchronized updates.

**Remark 1.** *The choice of $\beta$ in Theorem 1 is an arbitrary option that implies the rate in the theorem statement. The convergence proof may hold for any choices of $\beta$, such that $\beta K = \mathcal{O}(1)$.*

**Remark 2.** *In our analysis for Theorem 1, we considered bounded population diversity in Assumption 4. One can see that by relaxing this assumption to a stronger variant*

$$\max_{i\in[n]}\sup_{w\in\mathbb{R}^d}\|\nabla f_i(w) - \nabla f(w)\|^2 \leq \gamma^2, \tag{9}$$

*i.e., uniformly bounded heterogeneity[5], $n\gamma^2$ can be replaced with $\gamma^2$ in the third term of the rate.*

## 4   Conclusion

This paper studied the convergence properties of asynchronous federated learning via secure buffered aggregation. By removing the boundedness assumption on the gradient norms, we presented a novel analysis of the convergence of FedBuff algorithm, where we showed a sublinear convergence rate of $\mathcal{O}(\epsilon^2) + \mathcal{O}(\tau^2\epsilon)$ to an $\epsilon$-first-order stationary solution. We also discussed the dependence of this rate on the batch size, stochasticity variance, data heterogeneity, and maximum delays. We leave the privacy analysis of Fed-Buff with gradient clipping and noise addition to future studies. Also, the communication complexity of this method and the extensions to decentralized setups remain for future work.

## Acknowledgments and Disclosure of Funding

This work was partially funded by ARPA-H Strategic Initiative Seed Fund #916012. Part of this material is based upon work supported by the National Science Foundation under Grants #2211815 and #2213568.

---

[5]This stronger assumption is considered in the analysis of some related works such as [4][Assumption 3] and [22][6.1.1 Assumptions and Preliminaries, (vii)])

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

# A  Proof of Theorem 1

*Proof of Theorem 1.* Before proceeding with the proof, let us state some inequalities. For any set of $m$ vectors $\{w_i\}_{i=1}^m$ such that $w_i \in \mathbb{R}^d$, and a constant $\alpha > 0$, the following properties hold: for all $i, j \in [m]$:

$$\|w_i + w_j\|^2 \leq (1+\alpha)\|w_i\|^2 + (1+\alpha^{-1})\|w_j\|^2, \tag{10a}$$

$$\|w_i + w_j\| \leq \|w_i\| + \|w_j\|, \tag{10b}$$

$$2\langle w_i, w_j \rangle \leq \alpha\|w_i\|^2 + \alpha^{-1}\|w_j\|^2, \tag{10c}$$

$$\left\|\sum_{i=1}^m w_i\right\|^2 \leq m\left(\sum_{i=1}^m \|w_i\|^2\right), \tag{10d}$$

$$(1-\alpha)\left(1 + \frac{\alpha}{2}\right) \leq 1 - \frac{\alpha}{2}, \tag{10e}$$

$$(1-\alpha)\left(1 + \frac{2}{\alpha}\right) \leq \frac{2}{\alpha}. \tag{10f}$$

For simplicity, let us denote $\tilde{\nabla} f_i(w) = \nabla \tilde{f}_i(w, \mathcal{D}_i)$. Therefore, at round $t$, the server updates its parameter by receiving $\overline{\Delta}^t$, as follows:

$$w^{t+1} = w^t - \beta \overline{\Delta}^t \tag{11}$$

$$= w^t - \beta \sum_{k=0}^{K-1} \Delta_{i_{t,k}} \tag{12}$$

$$= w^t - \eta\beta \sum_{k=0}^{K-1} \sum_{q=0}^{Q-1} \tilde{\nabla} f_{i_{t,k}}\left(w_{i_{t,k},q}^{\tau_{i_{t,k}}^t}\right). \tag{13}$$

Due to Assumption 2, we can infer that $f$ is $L$-smooth, thus

$$f\left(w^{t+1}\right) \overset{(4)}{\leq} f(w^t) - \eta\beta \underbrace{\left\langle \nabla f(w^t), \sum_{k=0}^{K-1} \sum_{q=0}^{Q-1} \tilde{\nabla} f_{i_{t,k}}\left(w_{i_{t,k},q}^{\tau_{i_{t,k}}^t}\right)\right\rangle}_{=:S_1} \tag{14}$$

$$+ \frac{L\eta^2\beta^2}{2} \underbrace{\left\|\sum_{k=0}^{K-1} \sum_{q=0}^{Q-1} \tilde{\nabla} f_{i_{t,k}}\left(w_{i_{t,k},q}^{\tau_{i_{t,k}}^t}\right)\right\|^2}_{=:S_2} \tag{15}$$

First, we provide a lower bound on term $S_1$ in (14). Prior to show the bound, let us denote $\tilde{g}_i^t = \sum_{q=0}^{Q-1} \tilde{\nabla} f_i\left(w_{i,q}^{\tau_i^t}\right)$, $\tilde{g}^t = \frac{1}{n} \sum_{i=1}^{n} \tilde{g}_i^t$, $g_i^t = \sum_{q=0}^{Q-1} \nabla f_i\left(w_{i,q}^{\tau_i^t}\right)$, and $g^t = \frac{1}{n} \sum_{i=1}^{n} g_i^t$. Therefore,

$$\mathbb{E}\left[S_1\right] = \mathbb{E}\left[\mathbb{E}_{i_{t,k}}\left\langle \nabla f(w^t), \sum_{k=0}^{K-1}\sum_{q=0}^{Q-1} \tilde{\nabla} f_{i_{t,k}}\left(w_{i_{t,k},q}^{\tau_{i_{t,k}}^t}\right)\right\rangle\right] \tag{16}$$

$$= \mathbb{E}\left[\left\langle \nabla f(w^t), \frac{1}{n}\sum_{i=1}^{n}\sum_{k=0}^{K-1} \tilde{g}_i^t\right\rangle\right] \tag{17}$$

$$= \mathbb{E}\left\langle \nabla f(w^t), \frac{1}{n}\sum_{i=1}^{n}\sum_{k=0}^{K-1} \mathbb{E}_{p_i}\left[\tilde{g}_i^t\right]\right\rangle = \mathbb{E}\left\langle \nabla f(w^t), \frac{K}{n}\sum_{i=1}^{n} g_i^t\right\rangle \tag{18}$$

$$= KQ\,\mathbb{E}\left\|\nabla f(w^t)\right\|^2 + K\left[\mathbb{E}\left\langle \nabla f(w^t), g^t - Q\nabla f(w^t)\right\rangle\right] \tag{19}$$

$$\overset{(10c)}{\geq} KQ\,\mathbb{E}\left\|\nabla f(w^t)\right\|^2 - \frac{K}{2}\mathbb{E}\left\|\nabla f(w^t)\right\|^2 - \frac{K}{2}\mathbb{E}\left\|g^t - Q\nabla f(w^t)\right\|^2 \tag{20}$$

$$= \frac{K(2Q-1)}{2}\mathbb{E}\left\|\nabla f(w^t)\right\|^2 - \frac{K}{2}\mathbb{E}\left\|g^t - Q\nabla f(w^t)\right\|^2. \tag{21}$$

Moreover, the following holds for $S_2$ in (14):

$$\mathbb{E}\left[S_2\right] = \mathbb{E}\left[\mathbb{E}_{i_{t,k}}\left\|\sum_{k=0}^{K-1}\sum_{q=0}^{Q-1} \tilde{\nabla} f_{i_{t,k}}\left(w_{i_{t,k},q}^{\tau_{i_{t,k}}^t}\right)\right\|^2\right] \tag{22}$$

$$= \frac{1}{n}\mathbb{E}\left[\sum_{i=1}^{n}\left\|\sum_{k=0}^{K-1}\sum_{q=0}^{Q-1} \tilde{\nabla} f_i\left(w_{i,q}^{\tau_i^t}\right)\right\|^2\right] \tag{23}$$

$$= \frac{K^2}{n}\sum_{i=1}^{n}\mathbb{E}\left\|\sum_{q=0}^{Q-1} \tilde{\nabla} f_i\left(w_{i,q}^{\tau_i^t}\right)\right\|^2 = \frac{K^2}{n}\sum_{i=1}^{n}\mathbb{E}\left\|\tilde{g}_i^t\right\|^2. \tag{24}$$

Now, according to (14), (21), and (22), we have:

$$\mathbb{E}f\left(w^{t+1}\right) \leq \mathbb{E}f(w^t) - \frac{\eta\beta K(2Q-1)}{2}\mathbb{E}\left\|\nabla f(w^t)\right\|^2 \tag{25}$$

$$+ \frac{\eta\beta K}{2}\mathbb{E}\underbrace{\left\|g^t - Q\nabla f(w^t)\right\|^2}_{=:S_3} + \frac{L\eta^2\beta^2 K^2}{2n}\mathbb{E}\underbrace{\left[\sum_{i=1}^{n}\left\|\tilde{g}_i^t\right\|^2\right]}_{=:S_4}, \tag{26}$$

where we bound $S_3, S_4$ as follows:

$$S_3 = \left\|\frac{1}{n}\sum_{i=1}^{n}\left(g_i^t - Q\nabla f_i(w^t)\right)\right\|^2 \overset{(10d)}{\leq} \frac{1}{n}\sum_{i=1}^{n}\left\|g_i^t - Q\nabla f_i(w^t)\right\|^2 \tag{27}$$

$$= \frac{1}{n}\sum_{i=1}^{n}\left\|\sum_{q=0}^{Q-1} \nabla f_i\left(w_{i,q}^{\tau_i^t}\right) - Q\nabla f_i(w^t)\right\|^2 \tag{28}$$

$$= \frac{1}{n}\sum_{i=1}^{n}\left\|\sum_{q=0}^{Q-1}\left[\nabla f_i\left(w_{i,q}^{\tau_i^t}\right) - \nabla f_i(w^t)\right]\right\|^2 \tag{29}$$

$$\overset{(10d)}{\leq} \frac{Q}{n}\sum_{i=1}^{n}\sum_{q=0}^{Q-1}\left\|\nabla f_i\left(w_{i,q}^{\tau_i^t}\right) - \nabla f_i(w^t)\right\|^2, \tag{30}$$

$$S_4 = \sum_{i=1}^{n} \left\| \sum_{q=0}^{Q-1} \tilde{\nabla} f_i \left( w_{i,q}^{\tau_i^t} \right) \right\|^2 \tag{31}$$

$$\overset{(10d)}{\leq} Q \sum_{i=1}^{n} \sum_{q=0}^{Q-1} \left\| \tilde{\nabla} f_i \left( w_{i,q}^{\tau_i^t} \right) \right\|^2 \tag{32}$$

$$= Q \sum_{i=1}^{n} \sum_{q=0}^{Q-1} \left\| \tilde{\nabla} f_i \left( w_{i,q}^{\tau_i^t} \right) - \nabla f_i \left( w_{i,q}^{\tau_i^t} \right) + \nabla f_i \left( w_{i,q}^{\tau_i^t} \right) - \nabla f_i \left( w^t \right) \right.$$
$$\left. + \nabla f_i \left( w^t \right) - \nabla f \left( w^t \right) + \nabla f \left( w^t \right) \right\|^2 \tag{33}$$

$$\overset{(10d)}{\leq} 4Q \sum_{i=1}^{n} \sum_{q=0}^{Q-1} \left[ \left\| \tilde{\nabla} f_i \left( w_{i,q}^{\tau_i^t} \right) - \nabla f_i \left( w_{i,q}^{\tau_i^t} \right) \right\|^2 + \left\| \nabla f_i \left( w_{i,q}^{\tau_i^t} \right) - \nabla f_i \left( w^t \right) \right\|^2 \right.$$
$$\left. + \left\| \nabla f_i \left( w^t \right) - \nabla f \left( w^t \right) \right\|^2 + \left\| \nabla f \left( w^t \right) \right\|^2 \right] \Rightarrow \tag{34}$$

$$\mathbb{E}[S_4] \overset{(34)}{\leq} 4Q \sum_{i=1}^{n} \sum_{q=0}^{Q-1} \mathbb{E}_{p_i} \left[ \left\| \tilde{\nabla} f_i \left( w_{i,q}^{\tau_i^t} \right) - \nabla f_i \left( w_{i,q}^{\tau_i^t} \right) \right\|^2 \right] \tag{35}$$

$$+ 4Q \sum_{i=1}^{n} \sum_{q=0}^{Q-1} \mathbb{E} \left\| \nabla f_i \left( w_{i,q}^{\tau_i^t} \right) - \nabla f_i \left( w^t \right) \right\|^2 \tag{36}$$

$$+ 4Q \sum_{i=1}^{n} \sum_{q=0}^{Q-1} \mathbb{E} \left\| \nabla f_i \left( w^t \right) - \nabla f \left( w^t \right) \right\|^2 \tag{37}$$

$$+ 4Q \sum_{i=1}^{n} \sum_{q=0}^{Q-1} \mathbb{E} \left\| \nabla f \left( w^t \right) \right\|^2 \tag{38}$$

$$\overset{(7),(9)}{\leq} 4nQ^2 \left[ \hat{\sigma}^2 + \gamma^2 + \mathbb{E} \left\| \nabla f \left( w^t \right) \right\|^2 \right] \tag{39}$$

$$+ 4Q \sum_{i=1}^{n} \sum_{q=0}^{Q-1} \mathbb{E} \left\| \nabla f_i \left( w_{i,q}^{\tau_i^t} \right) - \nabla f_i(w^t) \right\|^2. \tag{40}$$

Therefore, due to (25)-(30), and (35)-(40), we have

$$\mathbb{E} f \left( w^{t+1} \right) \leq \mathbb{E} f(w^t) - \left[ \frac{\eta \beta K (2Q-1)}{2} - 2\eta^2 L \beta^2 K^2 Q^2 \right] \mathbb{E} \left\| \nabla f(w^t) \right\|^2 \tag{41}$$

$$+ \left[ \frac{\eta \beta K Q}{2n} + \frac{2\eta^2 \beta^2 K^2 Q L}{n} \right] \sum_{i=1}^{n} \sum_{q=0}^{Q-1} \mathbb{E} \left\| \nabla f_i \left( w_{i,q}^{\tau_i^t} \right) - \nabla f_i(w^t) \right\|^2 \tag{42}$$

$$+ 2\eta^2 L \beta^2 K^2 Q^2 \hat{\sigma}^2 + 2\eta^2 L \beta^2 K^2 Q^2 \gamma^2 \tag{43}$$

$$\overset{(4)}{\leq} \mathbb{E} f(w^t) - \left[ \frac{\eta \beta K (2Q-1)}{2} - 2\eta^2 L \beta^2 K^2 Q^2 \right] \mathbb{E} \left\| \nabla f(w^t) \right\|^2 \tag{44}$$

$$+ \frac{\eta \beta K Q L^2 \left( 1 + 4\eta \beta K L \right)}{2n} \sum_{i=1}^{n} \sum_{q=0}^{Q-1} \mathbb{E} \underbrace{\left\| w_{i,q}^{\tau_i^t} - w^t \right\|^2}_{=:S_5} \tag{45}$$

$$+ 2\eta^2 L \beta^2 K^2 Q^2 \hat{\sigma}^2 + 2\eta^2 L \beta^2 K^2 Q^2 \gamma^2. \tag{46}$$

Hence, it is sufficient to bound $S_5$ in (44),

$$S_5 = \left\| w^t - w_{i,q}^{\tau_i^t} \right\|^2 \tag{47}$$

$$= \left\| \sum_{s=\tau_i^t}^{t-1} \left( w^{s+1} - w^s \right) + w^{\tau_i^t} - w_{i,q}^{\tau_i^t} \right\|^2 \tag{48}$$

$$\overset{(10a)}{\leq} \left( 1 + \frac{1}{\beta^2 K^2} \right) \left\| \sum_{s=\tau_i^t}^{t-1} \left( w^{s+1} - w^s \right) \right\|^2 + \left( 1 + \beta^2 K^2 \right) \left\| w^{\tau_i^t} - w_{i,q}^{\tau_i^t} \right\|^2 \tag{49}$$

$$\overset{(10d)}{\leq} \left( t - \tau_i^t \right) \left( 1 + \frac{1}{\beta^2 K^2} \right) \left[ \sum_{s=\tau_i^t}^{t-1} \left\| w^{s+1} - w^s \right\|^2 \right] + \left( 1 + \beta^2 K^2 \right) \left\| w^{\tau_i^t} - w_{i,q}^{\tau_i^t} \right\|^2 \tag{50}$$

$$\overset{(3)}{\leq} \tau \left( 1 + \frac{1}{\beta^2 K^2} \right) \left[ \sum_{s=t-\tau}^{t-1} \underbrace{\left\| w^{s+1} - w^s \right\|^2}_{=:S_7} \right] + \left( 1 + \beta^2 K^2 \right) \underbrace{\left\| w^{\tau_i^t} - w_{i,q}^{\tau_i^t} \right\|^2}_{=:S_6}. \tag{51}$$

Now, we show a bound on the evolution of local updates at an arbitrary round $s \geq 0$, i.e., the distance between $w_{i,q}^s$ and $w^s$, which we will use to provide a bound on $S_7$.

$$\mathbb{E} \left\| w_{i,q}^s - w^s \right\|^2 = \mathbb{E} \left\| w_{i,q-1}^s - \eta \tilde{\nabla} f_i \left( w_{i,q-1}^s \right) - w^s \right\|^2 \tag{52}$$

$$= \mathbb{E} \Big\| w_{i,q-1}^s - w^s - \eta \nabla f \left( w^s \right)$$
$$\qquad - \eta \tilde{\nabla} f_i \left( w_{i,q-1}^s \right) + \eta \nabla f_i \left( w_{i,q-1}^s \right)$$
$$\qquad - \eta \nabla f_i \left( w_{i,q-1}^s \right) + \eta \nabla f_i \left( w^s \right)$$
$$\qquad - \eta \nabla f_i \left( w^s \right) + \eta \nabla f \left( w^s \right) \Big\|^2 \tag{53}$$

$$\overset{(10a)}{\leq} \left( 1 + \frac{1}{2Q} \right) \mathbb{E} \left\| w_{i,q-1}^s - w^s \right\|^2 \tag{54}$$

$$+ \quad 4(1+2Q)\eta^2 \mathbb{E} \Bigg[ \left\| \tilde{\nabla} f_i \left( w_{i,q-1}^s \right) - \nabla f_i \left( w_{i,q-1}^s \right) \right\|^2$$

$$+ \left\| \nabla f_i \left( w_{i,q-1}^s \right) - \nabla f_i \left( w^s \right) \right\|^2$$

$$+ \left\| \nabla f_i \left( w^s \right) - \nabla f \left( w^s \right) \right\|^2$$

$$+ \left\| \nabla f \left( w^s \right) \right\|^2 \Bigg] \tag{55}$$

$$\overset{(4),(7)}{\leq} \left( 1 + \frac{1}{2Q} \right) \mathbb{E} \left\| w_{i,q-1}^s - w^s \right\|^2 \tag{56}$$

$$+ \quad 4(1+2Q)\eta^2 \Bigg[ \hat{\sigma}^2 + L^2 \mathbb{E} \left\| w_{i,q-1}^s - w^s \right\|^2$$

$$+ \mathbb{E} \left\| \nabla f_i \left( w^s \right) - \nabla f \left( w^s \right) \right\|^2 + \mathbb{E} \left\| \nabla f \left( w^s \right) \right\|^2 \Bigg]. \tag{57}$$

Note that we can select stepsize $\eta \leq \frac{1}{4L(Q+1)}$ such that

$$\eta^2 \leq \frac{1}{16L^2(Q+1)^2} \leq \frac{1}{8L^2Q(2Q+1)} \Rightarrow 4(1+2Q)\eta^2 L^2 \leq \frac{1}{2Q}, \tag{58}$$

therefore, due to (52)-(57) and (58), we have:

$$\underbrace{\mathbb{E}\left\|w_{i,q}^s - w^s\right\|^2}_{:=P_{i,q}^s} \leq \underbrace{\left(1+\frac{1}{Q}\right)\mathbb{E}\left\|w_{i,q-1}^s - w^s\right\|^2}_{:=P_{i,q-1}^s} \tag{59}$$

$$+ \underbrace{4(1+2Q)\eta^2\left[\hat{\sigma}^2 + \mathbb{E}\left\|\nabla f_i\left(w^s\right) - \nabla f\left(w^s\right)\right\|^2 + \mathbb{E}\left\|\nabla f\left(w^s\right)\right\|^2\right]}_{:=R_i^s} \Rightarrow \tag{60}$$

$$P_{i,q}^s \leq \left(1+\frac{1}{Q}\right)P_{i,q-1}^s + R_i^s \tag{61}$$

$$= R_i^s \sum_{k=0}^{q-1}\left(1+\frac{1}{Q}\right)^k \leq R_i^s \sum_{k=0}^{Q-1}\left(1+\frac{1}{Q}\right)^k \tag{62}$$

$$= R_i^s \frac{\left(1+\frac{1}{Q}\right)^Q - 1}{\left(1+\frac{1}{Q}\right) - 1} = R_i^s Q\left[\left(1+\frac{1}{Q}\right)^Q - 1\right] \leq R_i^s Q(e-1) \leq 2R_i^s Q \Rightarrow \tag{63}$$

$$\mathbb{E}\left\|w_{i,q}^s - w^s\right\|^2 \leq 8Q(1+2Q)\eta^2\left[\hat{\sigma}^2 + \mathbb{E}\left\|\nabla f_i\left(w^s\right) - \nabla f\left(w^s\right)\right\|^2 + \mathbb{E}\left\|\nabla f\left(w^s\right)\right\|^2\right], \tag{64}$$

for all $q \in [Q]$. Again, note that according to Algorithm 2, we have:

$$w^{s+1} = w^s - \beta\sum_{k=0}^{K-1}\left[w_{i_s,0}^{\tau_{i_s}^s} - w_{i_s,Q}^{\tau_{i_s}^s}\right] = w^s - \beta\sum_{k=0}^{K-1}\left[w^{\tau_{i_s}^s} - w_{i_s,Q}^{\tau_{i_s}^s}\right] \Rightarrow \tag{65}$$

$$\mathbb{E}\left\|w^{s+1} - w^s\right\|^2 \leq \beta^2\,\mathbb{E}\left\|\sum_{k=0}^{K-1}\left[w^{\tau_{i_s}^s} - w_{i_s,Q}^{\tau_{i_s}^s}\right]\right\|^2 \tag{66}$$

$$\overset{(10d)}{\leq} \beta^2 K \sum_{k=0}^{K-1}\mathbb{E}\left\|w^{\tau_{i_s}^s} - w_{i_s,Q}^{\tau_{i_s}^s}\right\|^2 \tag{67}$$

$$= \beta^2 K^2\left[\mathbb{E}\left[\mathbb{E}_{i_s}\left\|w^{\tau_{i_s}^s} - w_{i_s,Q}^{\tau_{i_s}^s}\right\|^2\right]\right] \tag{68}$$

$$= \frac{\beta^2 K^2}{n}\sum_{j=1}^{n}\mathbb{E}\left\|w^{\tau_j^s} - w_{j,Q}^{\tau_j^s}\right\|^2 \tag{69}$$

$$\overset{(59)-(64)}{\leq} 8Q(1+2Q)\eta^2\beta^2 K^2\hat{\sigma}^2 \tag{70}$$

$$+ \frac{8Q(1+2Q)\eta^2\beta^2 K^2}{n}\sum_{j=1}^{n}\mathbb{E}\left\|\nabla f_j\left(w^{\tau_j^s}\right) - \nabla f\left(w^{\tau_j^s}\right)\right\|^2 \tag{71}$$

$$+ \frac{8Q(1+2Q)\eta^2\beta^2 K^2}{n}\sum_{j=1}^{n}\mathbb{E}\left\|\nabla f\left(w^{\tau_j^s}\right)\right\|^2. \tag{72}$$

Let $\phi = 8\eta^2 Q^2 (1+2Q)(1+\beta^2 K^2)$, then according to (47)-(72), we have

$$\frac{1}{n\phi} \sum_{i=1}^{n} \sum_{q=0}^{Q-1} \mathbb{E}[S_5] \leq \tau \left[ \sum_{s=t-\tau}^{t-1} \left\| w^{s+1} - w^s \right\|^2 \right] + \frac{1}{n} \sum_{i=1}^{n} \left\| w^{\tau_i^t} - w_{i,q}^{\tau_i^t} \right\|^2. \tag{73}$$

$$\leq \tau^2 \hat{\sigma}^2 + \frac{\tau}{n} \sum_{s=t-\tau}^{t-1} \sum_{j=1}^{n} \mathbb{E} \left\| \nabla f_j \left( w^{\tau_j^s} \right) - \nabla f \left( w^{\tau_j^s} \right) \right\|^2 \tag{74}$$

$$+ \frac{\tau}{n} \sum_{s=t-\tau}^{t-1} \sum_{j=1}^{n} \mathbb{E} \left\| \nabla f \left( w^{\tau_j^s} \right) \right\|^2 \tag{75}$$

$$+ \hat{\sigma}^2 + \frac{1}{n} \sum_{i=1}^{n} \mathbb{E} \left\| \nabla f_i \left( w^{\tau_i^t} \right) - \nabla f \left( w^{\tau_i^t} \right) \right\|^2 + \frac{1}{n} \sum_{i=1}^{n} \mathbb{E} \left\| \nabla f \left( w^{\tau_i^t} \right) \right\|^2. \tag{76}$$

Note that according to (3), we know that: $\tau_i^t \in \{t-\tau \ldots, t\}$, therefore:

$$\mathbb{E} \left\| \nabla f \left( w^{\tau_i^t} \right) \right\|^2 \leq \sum_{s=t-\tau}^{t} \mathbb{E} \left\| \nabla f \left( w^s \right) \right\|^2, \tag{77}$$

and similarly, for any $s \in \{t-\tau \ldots, t\}$ and $j \in [n]$,

$$\mathbb{E} \left\| \nabla f \left( w^{\tau_j^s} \right) \right\|^2 \leq \sum_{u=s-\tau}^{s} \mathbb{E} \left\| \nabla f \left( w^u \right) \right\|^2. \tag{78}$$

Moreover, we have:

$$\left\| \nabla f_j \left( w^{\tau_j^s} \right) - \nabla f \left( w^{\tau_j^s} \right) \right\|^2 \leq \sum_{i=1}^{n} \left\| \nabla f_i \left( w^{\tau_j^s} \right) - \nabla f \left( w^{\tau_j^s} \right) \right\|^2. \tag{79}$$

Therefore, due to (73)-(79), we have:

$$\frac{1}{n\phi} \sum_{i=1}^{n} \sum_{q=0}^{Q-1} \mathbb{E}[S_5] \overset{(77)-(79)}{\leq} \tau^2 \hat{\sigma}^2 + \tau^2 n \gamma^2 + \tau \sum_{s=t-\tau}^{t-1} \sum_{u=s-\tau}^{s} \mathbb{E} \left\| \nabla f \left( w^u \right) \right\|^2 \tag{80}$$

$$+ \hat{\sigma}^2 + n\gamma^2 + \sum_{s=t-\tau}^{t} \mathbb{E} \left\| \nabla f \left( w^s \right) \right\|^2 \tag{81}$$

$$= (1+\tau^2) \left[ \hat{\sigma}^2 + n\gamma^2 \right] + \tau \sum_{s=t-\tau}^{t-1} \sum_{u=s-\tau}^{s} \mathbb{E} \left\| \nabla f \left( w^u \right) \right\|^2 + \sum_{s=t-\tau}^{t} \mathbb{E} \left\| \nabla f \left( w^s \right) \right\|^2. \tag{82}$$

By combining (41)-(46) and (80)-(82), we have the following inequality:

$$\mathbb{E} f \left( w^{t+1} \right) \leq \mathbb{E} f(w^t) + 2\eta^2 L \beta^2 K^2 Q^2 \left[ \hat{\sigma}^2 + \gamma^2 \right] \tag{83}$$

$$- \frac{\eta \beta K}{2} \left[ (2Q-1) - 4\eta L \beta K Q^2 - Q L^2 (1+\tau^2)(1+4\eta \beta K L) \phi \right] \mathbb{E} \left\| \nabla f(w^t) \right\|^2 \tag{84}$$

$$+ \frac{\eta \beta K Q L^2 (1+4\eta \beta K L) \phi}{2} \sum_{s=t-\tau}^{t-1} \left[ \mathbb{E} \left\| \nabla f \left( w^s \right) \right\|^2 + \tau \sum_{u=s-\tau}^{s} \mathbb{E} \left\| \nabla f \left( w^u \right) \right\|^2 \right] \tag{85}$$

$$+ \frac{\eta \beta K Q L^2 (1+\tau^2)(1+4\eta \beta K L) \phi}{2} \left[ \hat{\sigma}^2 + n\gamma^2 \right] \tag{86}$$

$$\leq \mathbb{E} f(w^t) + 2\eta^2 L \beta^2 K^2 Q^2 \left[ \hat{\sigma}^2 + \gamma^2 \right] \tag{87}$$

$$- \frac{\eta \beta K Q}{2} \left[ 1 - 4\eta L \beta K Q - Q L^2 (1+\tau^2)(1+4\eta \beta K L) \phi \right] \mathbb{E} \left\| \nabla f(w^t) \right\|^2 \tag{88}$$

$$+ \frac{\eta \beta K Q L^2 (1+4\eta \beta K L) \phi}{2} \sum_{s=t-\tau}^{t-1} \left[ \mathbb{E} \left\| \nabla f \left( w^s \right) \right\|^2 + \tau \sum_{u=s-\tau}^{s} \mathbb{E} \left\| \nabla f \left( w^u \right) \right\|^2 \right] \tag{89}$$

$$+ \frac{\eta \beta K Q L^2 (1+\tau^2)(1+4\eta \beta K L) \phi}{2} \left[ \hat{\sigma}^2 + n\gamma^2 \right]. \tag{90}$$

Now, we can obtain the following inequality by rearranging the terms in (83)-(90):

$$\left[1 - 4\eta L\beta KQ - QL^2(1+\tau^2)\left(1+4\eta\beta KL\right)\phi\right]\mathbb{E}\left\|\nabla f(w^t)\right\|^2 \tag{91}$$

$$-L^2\left(1+4\eta\beta KL\right)\phi\sum_{s=t-\tau}^{t-1}\left[\mathbb{E}\left\|\nabla f\left(w^s\right)\right\|^2 + \tau\sum_{u=s-\tau}^{s}\mathbb{E}\left\|\nabla f\left(w^u\right)\right\|^2\right] \tag{92}$$

$$\leq \frac{2\left[\mathbb{E}f(w^t) - \mathbb{E}f\left(w^{t+1}\right)\right]}{\eta\beta KQ} \tag{93}$$

$$+ 4\eta\beta KQL\left[\hat\sigma^2 + \gamma^2\right] \tag{94}$$

$$+ 2L^2(1+\tau^2)\left(1+4\eta\beta KL\right)\phi\left[\hat\sigma^2 + n\gamma^2\right], \tag{95}$$

whereby mixing the terms in (92), we obtain:

$$\left[1 - 4\eta L\beta KQ - L^2(\tau^2+1)\left(1+4\eta\beta KL\right)\phi\right]\mathbb{E}\left\|\nabla f(w^t)\right\|^2 \tag{96}$$

$$-L^2\left(1+4\eta\beta KL\right)(\tau+1)\phi\sum_{s=t-\tau}^{t-1}\sum_{u=s-\tau}^{s}\mathbb{E}\left\|\nabla f\left(w^u\right)\right\|^2 \tag{97}$$

$$\leq \frac{2\left[\mathbb{E}f(w^t) - \mathbb{E}f\left(w^{t+1}\right)\right]}{\eta\beta KQ} \tag{98}$$

$$+ 4\eta\beta KQL\left[\hat\sigma^2 + \gamma^2\right] \tag{99}$$

$$+ 2L^2(1+\tau^2)\left(1+4\eta\beta KL\right)\phi\left[\hat\sigma^2 + n\gamma^2\right]. \tag{100}$$

Finally, we add (96)-(100), for $t = 0, 1, \ldots T-1$, and divide by $T$ to show that:

$$\left[1 - 4\eta L\beta KQ - L^2(\tau^2+1)\left(1+4\eta\beta KL\right)\phi \right.$$

$$\left. -L^2\left(1+4\eta\beta KL\right)\tau(\tau+1)^2\phi\right]\frac{\sum_{t=0}^{T-1}\mathbb{E}\left\|\nabla f(w^t)\right\|^2}{T} \tag{101}$$

$$\leq \frac{2\left[f(w^0) - \mathbb{E}f\left(w^T\right)\right]}{\eta\beta KQ} \tag{102}$$

$$+ 4\eta\beta KQL\left[\hat\sigma^2 + \gamma^2\right] \tag{103}$$

$$+ 2L^2(1+\tau^2)\left(1+4\eta\beta KL\right)\phi\left[\hat\sigma^2 + n\gamma^2\right]. \tag{104}$$

Let us fix $\beta = \frac{1}{K}$ and $\eta = \frac{1}{Q\sqrt{LT}}$. Thus, we know that the following inequality holds

$$\max\left\{4\eta\beta KLQ, \; L^2(\tau^2+1)(1+4\eta\beta KL)\phi, \; L^2\tau(\tau+1)^2(1+4\eta\beta KL)\phi\right\} \leq \frac{1}{4}, \tag{105}$$

for $T \geq 160L(Q+7)(\tau+1)^3$. Note that under this choices for $\eta$ and $\beta$, we also have $\eta \leq \frac{1}{4L(Q+1)}$, which we used in (58). Therefore, we can conclude the final result in Theorem 1 as follows:

$$\frac{1}{T}\sum_{t=0}^{T-1}\mathbb{E}\left\|\nabla f(w^t)\right\|^2 \leq \frac{8\sqrt{L}\left(f(w^0) - \mathbb{E}f\left(w^T\right)\right)}{\sqrt{T}} \tag{106}$$

$$+ \frac{16\sqrt{L}\left(\hat\sigma^2 + \gamma^2\right)}{\sqrt{T}} \tag{107}$$

$$+ \frac{320L(Q+1)(\tau^2+1)\left(\hat\sigma^2 + n\gamma^2\right)}{T}. \tag{108}$$

$$\square$$

