# OpenReview forum: "Unbounded Gradients in Federated Learning with Buffered Asynchronous Aggregation"
_NeurIPS.cc/2022/Workshop/Federated_Learning — FL-NeurIPS 2022 Poster_

### Official Review · Reviewer_esEU · 2022-10-01

## Summary
The paper presents a theoretical analysis of FedBuff without bounded gradient assumption.

## Strengths
* The paper removes the bounded gradient assumption from asynchronous FL analysis, this is the paper to my knowledge that does this. Removing bounded gradient assumption for async FL is a non-trivial a contribution.

## Weaknesses
* While the paper removes the bounded gradient assumption, it still relies on the bounded staleness. Koloskova et al provided tighter convergence rate for the homogeneous setup  (different setting from FedBuff) with by removing the dependence on the maximum delay without assuming bounded gradient.

Koloskova, Anastasia, Sebastian U. Stich, and Martin Jaggi. "Sharper Convergence Guarantees for Asynchronous SGD for Distributed and Federated Learning." arXiv preprint arXiv:2206.08307 (2022).

---

### Official Review · Reviewer_s7eh · 2022-10-19
**Improved analysis for FedBuff**

======Summary======
This paper provides an improved analysis for FedBuff, an asynchronous federated optimization algorithm. Specifically, in the original analysis of FedBuff, people assumed that the gradient norm is uniformly upper bounded. This paper relaxes this assumption by assuming only the gradient variance is bounded. This improvement makes sense and may be useful for future researchers. But compared to the FedBuff paper, the contributions of this paper is a bit limited. But for a workshop paper, it may be enough.

=======Comments=====
- It would be better if the authors can explain what are the benefits of having an improved analysis. What kind of new insights we can get?
- From the theoretical results, the dependence on the local steps Q is just linear. This is a bit surprising. As in the analysis of FedAvg, the dependence on Q should be quadratic. If this is true, that means asynchronous algorithm is more robust to the increase of local steps. The authors should provide more justifications on this difference.

---

### Official Review · Reviewer_9tw1 · 2022-10-19
**This paper is beyond my expertise.**

This paper is beyond my expertise.

---

### Decision · Program_Chairs · 2022-10-20

Accept (Poster)